Corrected: Publisher correction

# Evolutionary adaptations to new environments generally reverse plastic phenotypic changes

Wei-Chin Ho[1,2] & Jianzhi Zhang [1]

Organismal adaptation to a new environment may start with plastic phenotypic changes followed by genetic changes, but whether the plastic changes are stepping stones to genetic adaptation is debated. Here we address this question by investigating gene expression and metabolic flux changes in the two-phase adaptation process using transcriptomic data from multiple experimental evolution studies and computational metabolic network analysis, respectively. We discover that genetic changes more frequently reverse than reinforce plastic phenotypic changes in virtually every adaptation. Metabolic network analysis reveals that, even in the presence of plasticity, organismal fitness drops after environmental shifts, but largely recovers through subsequent evolution. Such fitness trajectories explain why plastic phenotypic changes are genetically compensated rather than strengthened. In conclusion, although phenotypic plasticity may serve as an emergency response to a new environment that is necessary for survival, it does not generally facilitate genetic adaptation by bringing the organismal phenotype closer to the new optimum.

[1] Department of Ecology and Evolutionary Biology, University of Michigan, Ann Arbor, MI 48109, USA. [2] Present address: Center for Mechanisms of Evolution, The Biodesign Institute, Arizona State University, Tempe, AZ 85287, USA. Correspondence and requests for materials should be addressed to J.Z. (email: jianzhi@umich.edu)

Phenotypic adaptation to a new environment can comprise two phases (Fig. 1a). In the first phase, the environmental shift induces phenotypic changes without mutation; such changes are referred to as plastic changes (PCs) irrespective of their fitness effects. After the first phase, there can be a second phase during which phenotypes are altered by mutations that accumulate during adaptive evolution. While most past evolutionary studies focused on the second phase, recent years have seen a growth in the argument for the importance of the first phase in adaptation[1–9]. Specifically, it is suggested that plastic phenotypic changes are often necessary for organismal survival in a new environment[10,11], which is essential because no adaptive evolution is possible if the environmental shift kills all individuals. Furthermore, it is suggested that genetic adaptations in the second phase are eased by the PCs in the first phase[1,2]. For example, plasticity can move the phenotypic value of an organism closer to the adapted state in the new environment and serve as a stepping stone to adaptation[7] (Fig. 1b). While some case studies appear to support this latter assertion[8,12,13], its general validity remains unclear[14]. Assessing the general validity is especially relevant because the school of extended evolutionary synthesis believes that plasticity is generally critical to adaptation and hence is requesting a major revision of the modern synthesis of evolutionary biology, where the role of plasticity in adaptation is thought to be largely neglected[1,2].

For a trait, its plastic phenotypic change induced by an environmental shift and the subsequent genetic change (GC) during the adaptation to the new environment could be in the same direction toward the optimal phenotypic value in the new environment. In this case, the PC is reinforced by the adaptive GC and hence is considered adaptive[14,15] (Fig. 1b). The PC and the subsequent GC could also be in opposite directions. In this case, the PC is reversed by the adaptive GC and is thus commonly considered non-adaptive[14,15] (Fig. 1c). Because the PCs and GCs are either in the same direction or opposite directions, the null expectation under no specific relationship between the two changes is that reinforcement and reversion are equally probable. The hypothesis that plasticity generally facilitates adaptation would be supported if reinforcement is more prevalent than reversion in a large sample of traits during a large number of adaptations; otherwise, the hypothesis is refuted. Thus, a test of the hypothesis can be performed by phenotyping and comparing adapted organisms in the original and new environments as well as the organisms right after the environmental shift (i.e., after PCs but before GCs). Early tests used morphological, physiological, or behavioral traits, but the number of traits examined was small and the results varied among studies[14]. Recent tests with transcriptome data suggested that gene expression level reversion is more prevalent than reinforcement during experimental evolution[15–18]. Although the number of traits is large in these recent

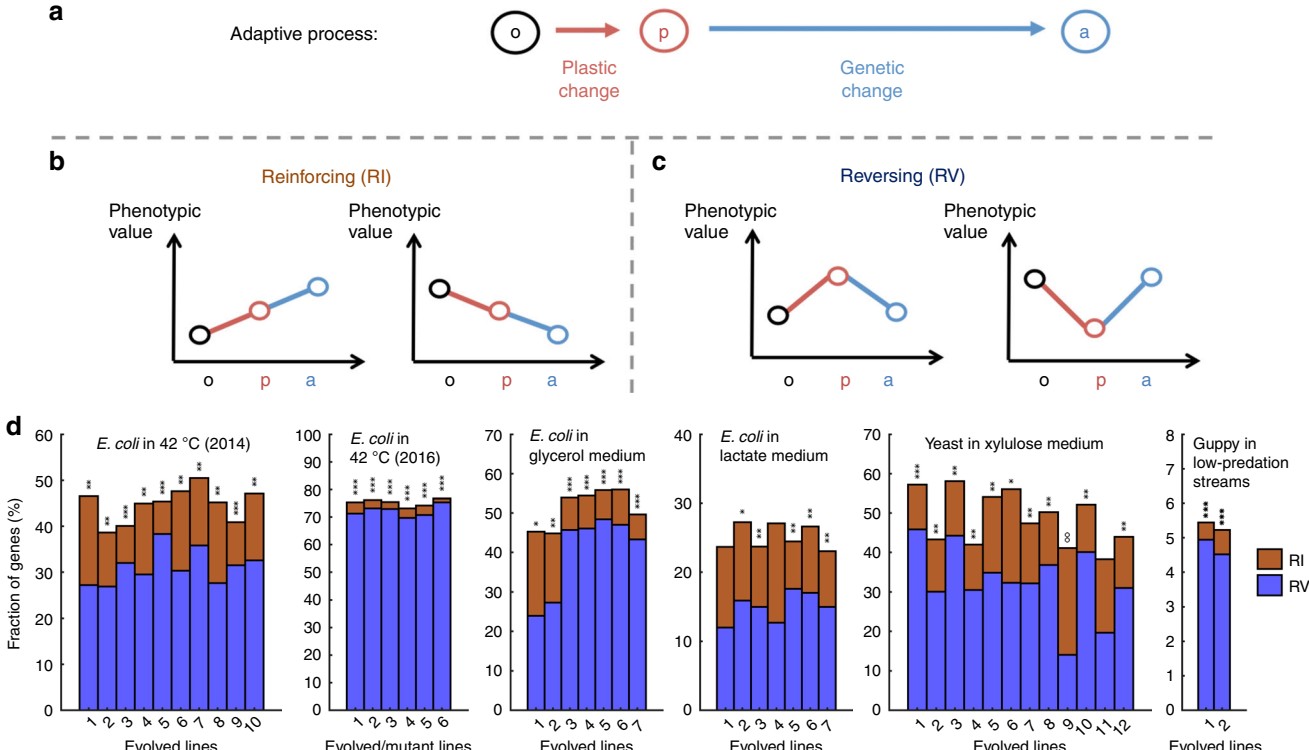

**Fig. 1** Gene expression changes in experimental evolution. **a** Phenotypic adaptation is studied by comparing the phenotypic values of a trait at three stages: ancestral organisms adapted to the original environment measured in the original environment (stage o); ancestral organisms measured in the new environment (stage p); and evolved organisms adapted to the new environment measured in the new environment (stage a). Plastic changes refer to changes from stage o to p, while genetic changes refer to changes from stage p to a. **b** A pair of plastic and genetic phenotypic changes of a trait are said to be reinforcing if both are larger than a preset cutoff and are in the same direction. **c** A pair of plastic and genetic phenotypic changes of a trait are said to be reversing if both are larger than a preset cutoff but are in opposite directions. **d** Fractions of genes with reinforcing ($C_{RI}$) and reversing ($C_{RV}$) expression changes, respectively, in experimental evolution. Organisms as well as the new environments to which the organisms were adapting to are indicated. Each bar represents an adaptation. The equality in the fraction of reinforcing and reversing genes in each adaptation is tested by a two-tailed binomial test. When $C_{RV} > C_{RI}$, P-values are indicated as follows: *P < 0.05; **P < $10^{-10}$; ***P < $10^{-100}$; when $C_{RV} < C_{RI}$, P-values are indicated as follows: °P < 0.05; °°P < $10^{-10}$; °°°P < $10^{-100}$

studies, their analyses vary, rendering the interpretation and among-study comparison difficult. We thus reanalyze using a uniform method the transcriptome data from these studies as well as those from another study that did not address the role of plasticity in adaptation[19].

More importantly, five considerations prompt us to expand the analysis from gene expression levels to another set of traits— metabolic fluxes. First, it is desirable to test the hypothesis across diverse environmental shifts, but experimental evolution studies with transcriptome data are currently limited in this aspect. By contrast, fluxes in well-annotated metabolic networks can be computationally predicted with reasonably high accuracy under a wide range of environments[20,21]. Second, it is necessary to examine if the finding from gene expression traits applies to other phenotypic traits. Third, organisms acquired at the end of experimental evolution are usually partially rather than fully adapted to the new environment, making the distinction between reinforcement and reversion less certain. Fourth, in experimental evolution, it is unknown whether an observed gene expression change is beneficial, neutral, or even deleterious. For example, an expression change accompanying organismal adaptation could be responsible for, a result from, or even unrelated to the fitness gain. Some authors assume that expression changes observed in replicate experiments are beneficial[15], but it is also possible that they are consequences of adaptation and have positive, zero, or negative fitness effects. Thus, not all expression changes observed in experimental evolution are relevant to the hypothesis that plasticity is a stepping stone to genetic adaptation. By contrast, in the metabolic network analysis, all flux changes observed in the maximization of fitness are required and therefore are beneficial. It has been shown, for instance, that upon the maximization of fitness, alteration of any non-zero flux would be deleterious[22]. Last and most importantly, because the regulatory and evolutionary mechanisms of gene expression changes are not well understood, it would be difficult to discern the mechanistic basis of expression level reinforcement or reversion. By contrast, patterns of computationally predicted flux changes can be understood mechanistically by the metabolic model used in the prediction. We thus test whether plasticity facilitates adaptation by computational metabolic flux analysis of the model bacterium *Escherichia coli*. Our analyses of transcriptome and fluxome changes in numerous adaptations consistently show that phenotypic reinforcement is not only no more but actually less prevalent than reversion, indicating that plasticity is not a stepping stone to genetic adaptation. More importantly, we uncover the underlying cause of the preponderance of phenotypic reversion.

## Results

**Prevalence of expression reversion in experimental evolution**. We identified five studies that conducted six different adaptation experiments and collected transcriptome data suiting our analysis. These six experiments included 10 replicates of *E. coli* adapting to a high-temperature environment[17], 6 replicates of another strain of *E. coli* adapting to a high-temperature environment[18], 7 replicates of *E. coli* adapting to a glycerol medium[16], 7 replicates of *E. coli* adapting to a lactate medium[16], 1 replicate each of 12 different yeast (*Saccharomyces cerevisiae*) strains adapting to an xylulose medium[19], and 2 replicates of guppies (*Poecilia reticulata*) adapting to a low-predation environment[15]. In total, we analyzed 44 cases of adaptation.

In each case, transcriptome data were respectively collected for the organisms in the original environment (o for the original stage), shortly after their exposure to the new environment (p for the plastic stage), and at the conclusion of the experimental

evolution in the new environment (a for the adapted stage; Fig. 1a). Note that the time between o and p is so short that no newly arisen allele is expected to have reached an appreciable frequency in stage p to impact the average phenotype of the population. The expression level of each gene is treated as a trait. Let the expression levels of a gene at the o, p, and a stages be $L_o$, $L_p$, and $L_a$, respectively. In each experiment, we first identified genes with appreciable PCs in expression level by requiring PC = | $L_p$–$L_o$| to be greater than a preset cutoff. We also identified genes with appreciable GCs in expression level by requiring GC = |$L_a$– $L_p$| to be greater than the same preset cutoff. For those genes showing both appreciable PCs and appreciable GCs, we ask whether the two changes are in the same direction (i.e., reinforcement) or opposite directions (i.e., reversion; Fig. 1b, c). We used 20% of the original gene expression level (i.e., $0.2L_o$) as the cutoff in the above analysis. The fraction of genes exhibiting expression level reinforcement ($C_{RI}$) is smaller than the fraction of genes exhibiting reversion ($C_{RV}$) in 42 of the 44 adaptations, and the difference between $C_{RI}$ and $C_{RV}$ is significant in 40 of these 42 cases (nominal $P < 0.05$; two-tailed binomial test; Fig. 1d). Among the two adaptations with $C_{RI} > C_{RV}$, their difference is significant in only one case (Fig. 1d). The general preponderance of expression level reversion (i.e., 42 of 44 cases) in adaptation is statistically significant ($P = 1.1 \times 10^{-10}$, two-tailed binomial test). The same trend is evident when the cutoff is altered to $0.05L_o$ (Supplementary Fig. 1a) or $0.5L_o$ (Supplementary Fig. 2a), suggesting that the above finding is robust to the cutoff choice. Clearly, the transcriptomic data do not support the hypothesis that plasticity generally facilitates genetic adaptation.

**Metabolic flux reversion in environmental adaptations**. To assess the generality of the above finding and understand its underlying cause, we expanded the comparison between phenotypic reinforcement and reversion to metabolic fluxes (see Introduction). Because our metabolic analysis is not meant to model the above experimental evolution or expression changes, the parameters used are unrelated to the experimental evolution. Specifically, we computationally predicted plastic and genetic flux changes during environmental adaptations using *i*AF1260, the reconstructed *E. coli* metabolic network[23]. We used flux balance analysis (FBA) to predict the optimized fluxes of fully adapted organisms in the original (stage o) and new (stage a) environments, respectively, under the assumption that the biomass production rate, a proxy for fitness, is maximized by natural selection[20]. FBA predictions match experimental measures reasonably well for organisms adapted to their environments[24–29] and are commonly used in the study of genotype–environment–phenotype relationships[22,27,29–37]. When predicting plastic flux changes upon environmental shifts (stage p), we employed minimization of metabolic adjustment (MOMA) instead of FBA because MOMA better recapitulates the immediate flux response to perturbations[21] (see Methods). We treated the flux of each reaction in the metabolic network as a trait, and modeled environmental shifts by altering the carbon source available to the network. There are 258 distinct exchange reactions in *i*AF1260, each transporting a different carbon source. We therefore examined 258 different single-carbon source environments.

We started the analysis by using glucose as the carbon source in the original environment, because this environment was the benchmark in *i*AF1260 construction[23]. We then considered the adaptations of *E. coli* to 257 new environments each with a different single-carbon source. We found that these new environments are naturally separated into two groups in the MOMA-predicted biomass production rate, a proxy for the

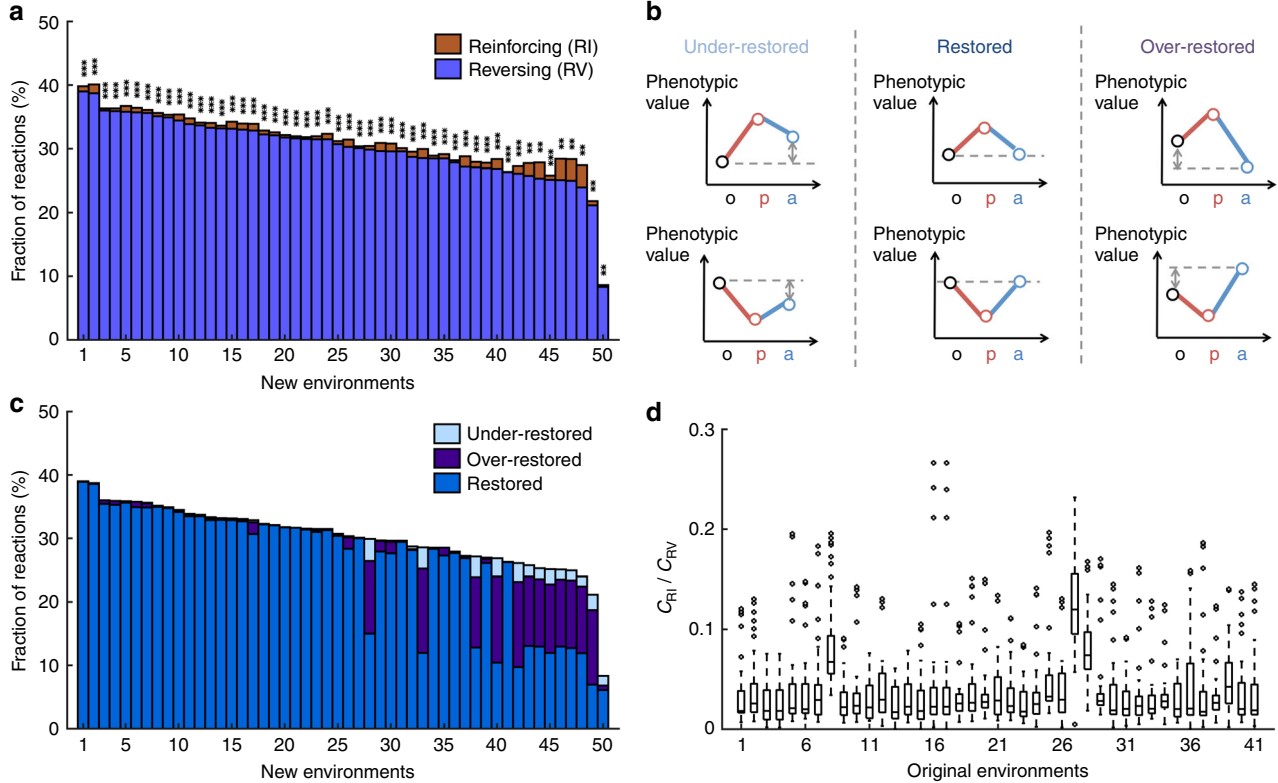

**Fig. 2** Predominance of flux reversion in the environmental adaptations of *E. coli*. **a** Fractions of reactions with reinforcing ($C_{RI}$) and reversing ($C_{RV}$) flux changes, respectively, in the adaptation from the glucose environment to each of 50 new environments. Each bar represents the adaptation to a new environment. The equality in the fraction of reinforcing and reversing reactions is tested by a two-tailed binomial test. When $C_{RV} > C_{RI}$, P-values are indicated as follows: *$P < 0.05$; **$P < 10^{-10}$; ***$P < 10^{-100}$; when $C_{RV} < C_{RI}$, P-values are indicated as follows: º$P < 0.05$; ºº$P < 10^{-10}$; ººº$P < 10^{-100}$. **b** Classification of reversion to three categories based on whether the phenotypic value in the original environment is under-restored, restored, or over-restored. **c** Fractions of the three categories of reversion in each of the 50 adaptations. **d** Fraction of reinforcing reactions relative to that of reversing reactions ($C_{RI}/C_{RV}$) in *E. coli* adaptations to at least 20 new environments from each of 41 original environments examined. The $C_{RI}/C_{RV}$ ratios for all adaptations from each original environment are presented in a box plot, where the lower and upper edges of a box represent the first ($qu_1$) and third ($qu_3$) quartiles, respectively, the horizontal line inside the box indicates the median (md), the whiskers extend to the most extreme values inside inner fences, md ± 1.5($qu_3 − qu_1$), and the circles represent values outside the inner fences (outliers)

fitness at stage p ($f_p$) (Supplementary Fig. 3). One group shows $f_p < 10^{-4}$, suggesting that *E. coli* is unlikely to sustain in these new environments. We therefore focused on the remaining 50 new environments with $f_p > 10^{-4}$, to which *E. coli* can presumably adapt (Supplementary Table 1).

Defining flux reinforcement and reversion and using the cutoff of $0.2L_o$ as in the transcriptome analysis, we found $C_{RV}$ to be significantly greater than $C_{RI}$ (nominal $P < 10^{-10}$, two-tailed binomial test) in each adaptation. The chance probability that all 50 adaptations show $C_{RV} > C_{RI}$ is $1.8 \times 10^{-15}$ (two-tailed binomial test; Fig. 2a), suggesting a general predominance of flux reversion. The mean and median $C_{RV}$ are 30.2% and 30.5%, respectively, while those for $C_{RI}$ are only 1.0% and 0.8%, respectively. The above trend holds when we alter the cutoff to $0.05L_o$ (Supplementary Fig. 1b) or $0.5L_o$ (Supplementary Fig. 2b). Because an FBA or MOMA problem may have multiple solutions, the order of the reactions in the stoichiometric matrix could affect the specific solution provided by the solver. Nevertheless, when we randomly shuffled the reaction order in iAF1260, the general pattern of $C_{RV} > C_{RI}$ is unaltered (Supplementary Fig. 4a). Because quadratic programming—required by MOMA—is harder to solve than linear programming used in FBA, $C_{RV}$ could have been overestimated compared with $C_{RI}$. To rectify this potential problem, we designed a quadratic programming-based

MOMA named "MOMA-b" and used it instead of FBA to predict fluxes at stage a (see Methods), but found that $C_{RV}$ still exceeds $C_{RI}$ (Supplementary Fig. 4b). Thus, this trend is not a technical artifact of the solver difference between MOMA and FBA.

**Flux reversion largely restores the original fluxes**. To examine whether the flux reversion during genetic adaptation restores the fluxes at stage o, we compared the total change TC = |$L_a − L_o$| with $0.2L_o$ for each reaction showing flux reversion, in each adaptation. If TC < $0.2L_o$, the flux is considered restored (Fig. 2b). Otherwise, we further compare PC with GC. If GC > PC, the flux is over-restored; otherwise, it is under-restored (Fig. 2b). Across the 50 adaptations, the mean fractions of reactions showing "restored", "over-restored", and "under-restored" flux reversion are 26.4%, 3.1%, and 0.7%, respectively, and the medians are 30.2%, 0.3%, and 0.1%, respectively (Fig. 2c). Clearly, flux reversion largely restores the fluxes at stage o.

**Predominance of flux reversion irrespective of the original environment**. To investigate the generality of our finding of the predominance of flux reversion, we also examined adaptations with a non-glucose original environment. For many original environments, however, only a few new environments are adaptable by the *E. coli* metabolic network. We thus focused on 41

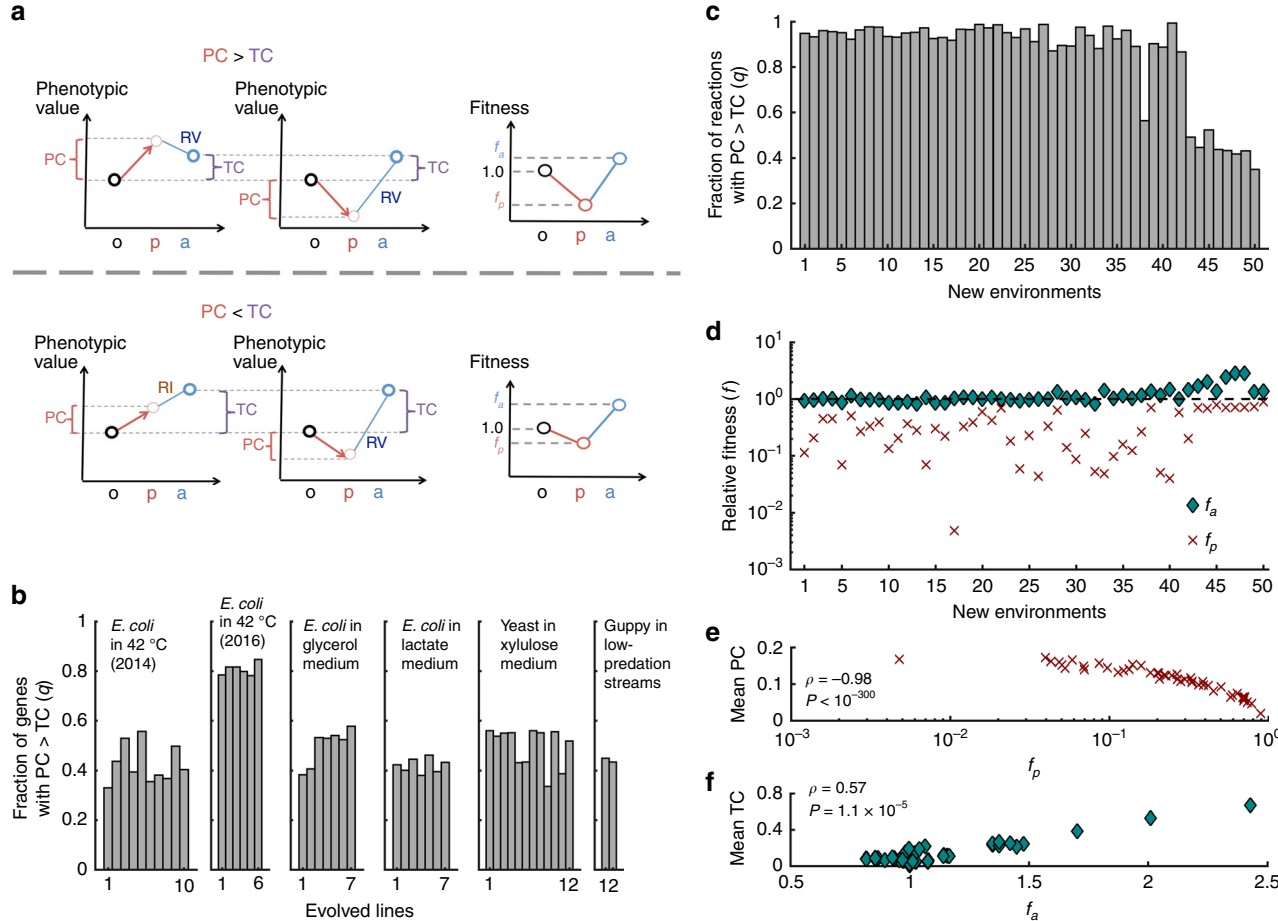

**Fig. 3** Cause of the preponderance of phenotypic reversion in adaptation. **a** Diagram illustrating the model. The upper part shows that if the plastic change (PC) is greater than the total change (TC), the genetic change (GC) must reverse the PC (the left and middle diagrams). One reason for PC > TC is that the fitness difference between organisms at stages o and p is greater than that between stages o and a (the right diagram). The lower part shows that if PC < TC, the GC either reinforces or reverses the PC (the left and middle diagrams). This may occur if the fitness difference between organisms at stages o and p is smaller than that between o and a (the right diagram) or if the phenotype is unassociated with fitness. **b** Fraction of genes showing expression PC > TC during each of 44 experimental evolutionary adaptations. **c** Fraction of reactions showing flux PC > TC during each of the *E. coli* metabolic adaptations from the glucose environment to the 50 new environments. **d** Fitness at stage p and that at stage a, relative to that at stage o, predicted by metabolic network analysis, for each of the 50 adaptations in **c**. The dotted line shows the fitness at stage o. **e** Mean PC across all fluxes negatively correlates with the relative fitness at stage p ($f_p$) among the 50 adaptations in **c**. **f** Mean TC across all fluxes positively correlates with the relative fitness at stage a ($f_a$) among the 50 adaptations in **c**

original environments (including the previously used glucose environment) that each has more than 20 adaptable (i.e., $f_p > 10^{-4}$) new environments (Supplementary Table 2). For each of these original environments, we calculated the $C_{RI}/C_{RV}$ ratio for each adaptable new environment, and found it to be typically lower than 0.1 (Fig. 2d). We then computed the median $C_{RI}/C_{RV}$ across all adaptable new environments from each original environment. Across the 41 original environments, the largest median $C_{RI}/C_{RV}$ is 0.11 and the median of median $C_{RI}/C_{RV}$ is only 0.02. Hence, regardless of the original environment, flux reversion is much more prevalent than reinforcement during genetic adaptations to new environments.

**Why phenotypic reversion is more frequent than reinforcement**. Our finding that phenotypic reinforcement is not only no more but actually much less common than reversion is unexpected and hence demands an explanation. The observation of this trend in both transcriptomic and fluxomic analyses suggests a general underlying mechanism, which we propose is the

occurrence of PC > TC. Geometrically, it is obvious that when PC > TC, the GC must reverse the PC (the left and middle diagrams in the top row in Fig. 3a). By contrast, when PC < TC, reversion and reinforcement are equally likely if no other bias exists (the left and middle diagrams in the bottom row in Fig. 3a). Let the probability of PC > TC be $q$ ($> 0$). $C_{RI}/C_{RV}$ is expected to be $[0.5(1 - q)]/[0.5(1 - q) + q] = (1 - q)/(1 + q) < 1$. In other words, as long as PC > TC for a few traits, reversion is expected to be more frequent than reinforcement (under no other bias).

To seek empirical evidence for the above explanation, for each of the 44 cases of experimental evolution, we calculated the fraction of genes whose expression changes satisfy PC > TC (Fig. 3b). The mean and median fractions are 0.51 and 0.48, respectively. Furthermore, after we remove all genes for which PC > TC, there is no longer an excess of reversion (Supplementary Fig. 5a), indicating the sufficiency of our explanation. Similarly, we computed the fraction of metabolic reactions showing PC > TC in the adaptation of the *E. coli* metabolic network from the glucose environment to each of the 50 new environments (Fig. 3c). The mean and median fractions are 0.85

and 0.93, respectively. Similarly, after the removal of reactions showing PC > TC, there is no general trend of more reversion than reinforcement across the 50 adaptations (Supplementary Fig. 5b). These transcriptome and fluxome results support that the excess of reversion relative to reinforcement is explainable by the occurrence of PC > TC for non-negligible fractions of traits.

Why does PC exceed TC for many traits? A likely reason is that PCs allow organisms to survive upon a sudden environmental shift but the fitness is much reduced compared with that in the original environment as well as that after the adaptation to the new environment. Thus, the overall physiological state of the organisms may be quite similar between the adapted stages in the original and new environments, but is much different in the low-fitness plastic stage right after the environmental shift. This may explain why PC exceeds $TC$ for many traits, regardless of whether the trait values are causes or consequences of the organismal fitness and physiology.

We found strong evidence for the above model by metabolic network analysis. First, using the predicted biomass production rate as a proxy for fitness, we compared the *E. coli* fitness at the plastic stage ($f_p$) and that after adaptation to a new environment ($f_a$), relative to that in the original glucose environment, for each of the adaptations to the 50 new environments. In all cases, $f_p < 1$ (Fig. 3d), confirming that environmental shifts cause fitness drops before genetic adaptation. We found that $f_a$ is typically close to 1, although in a few new environments it is much >1 (Fig. 3d). In a $\log_{10}$ scale, $f_p$ is more different from 1 than is $f_a$ in 43 of the 50

adaptations ($P = 1.0 \times 10^{-7}$; one-tailed binomial test). Second, our model assumes an association between flux changes and fitness changes[22]. Across the 50 adaptations from the glucose environment, there is a strong negative correlation between $f_p$ and mean PC (Spearman's $\rho = -0.98$, $P < 10^{-300}$; Fig. 3e). An opposite correlation exists between $f_a$ and mean TC ($\rho = 0.57$, $P = 1.1 \times 10^{-5}$; Fig. 3f). Together, our analyses demonstrate that the primary reason for a higher frequency of phenotypic reversion than reinforcement during adaptation is that in terms of fitness and associated phenotypes, organisms at stage p are more different than those at stage a, when compared with those at stage o.

**Phenotypic reversion in random metabolic networks**. The PCs and GCs in gene expression level and metabolic flux during adaptations depend, respectively, on the regulatory network and metabolic network of the species concerned. Because these networks result from billions of years of evolution, one wonders whether the predominance of phenotypic reversion is attributable to the evolutionary history of the species studied, especially the environments in which the species and its ancestors have been selected in the past, or an intrinsic property of any functional system. To address this question, we applied the same analysis to 500 functional random metabolic networks previously generated[22]. These networks were constructed from *i*AF1260 by swapping its reactions with randomly picked reactions from the universe of all metabolic reactions in Kyoto Encyclopedia of

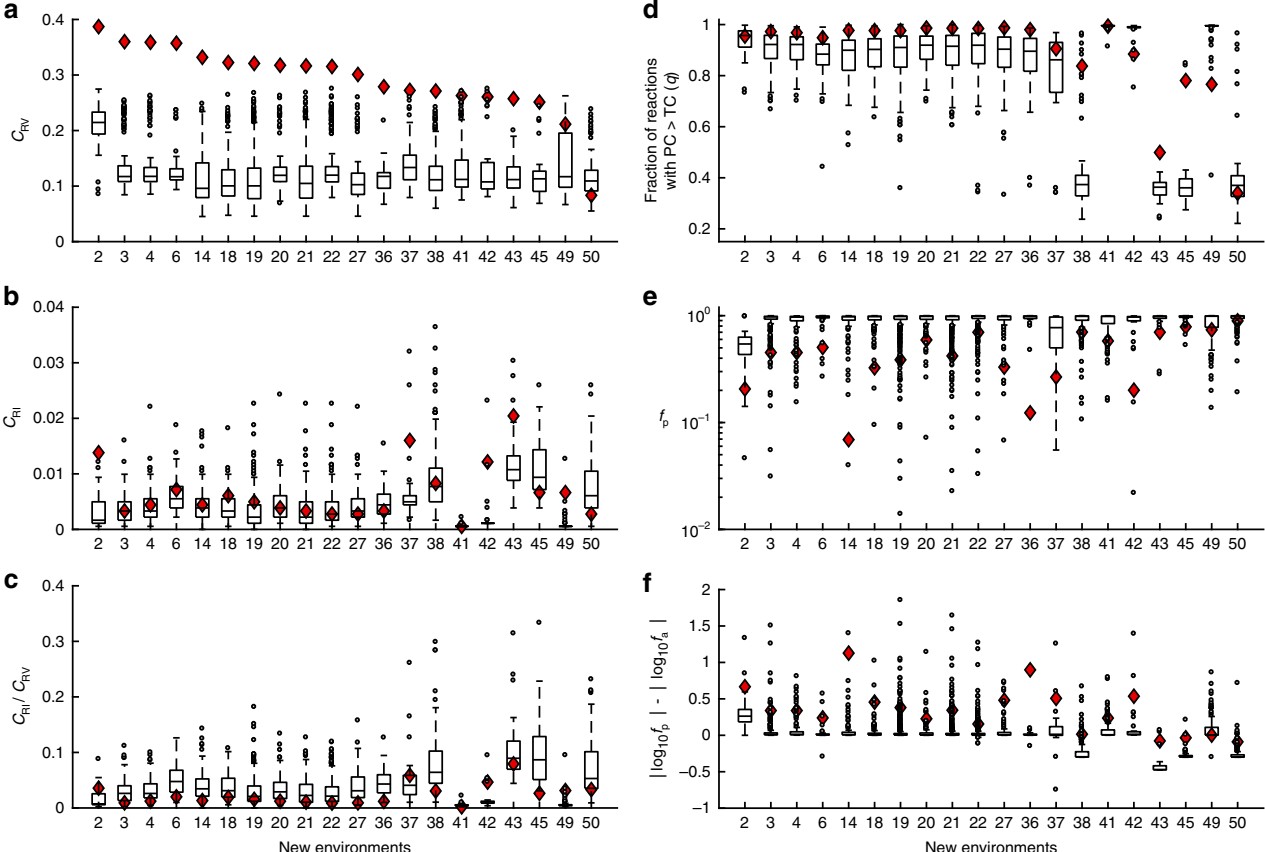

**Fig. 4** Predominance of flux reversion in random metabolic networks. Fractions of reactions showing flux reversion ($C_{RV}$) (**a**), fractions of reactions showing flux reinforcement ($C_{RI}$) (**b**), $C_{RI}/C_{RV}$ ratios (**c**), fraction of reactions showing PC > TC (**d**), $f_p$ (**e**), and $|\log_{10}f_p| - |\log_{10}f_a|$ (**f**) in the adaptations of random networks from the glucose environment to each of the 20 new environments examined. For each new environment, values estimated from different random networks are shown by a box plot, with symbols explained in the legend to Fig. 2d. The corresponding values for the *E. coli i*AF1260 network are shown by red diamonds

Genes and Genomes[38] as long as the network has a non-zero FBA-predicted fitness in the glucose environment upon each reaction swap[39].

Only 20 new environments that $iAF1260$ can adapt to (from the glucose environment) are adaptable by at least 20 of the 500 random networks. We thus analyzed the adaptations of random networks to each of these 20 new environments, with the glucose environment being the original environment. For each new environment, the median $C_{RV}$ of all random networks that can adapt to this environment is generally around 0.1 (box plots in Fig. 4a), with the median of median $C_{RV}$ being 0.11. By contrast, median $C_{RI}$ across random networks for a new environment is generally below 0.01 (box plots in Fig. 4b), with the median of median $C_{RI}$ being 0.0033. Median $C_{RI}/C_{RV}$ ratio across random networks for a new environment is generally below 0.05 (box plot in Fig. 4c), with the median of the median $C_{RI}/C_{RV}$ being 0.0033. Clearly, the predominance of flux reversion is also evident in functional random networks, suggesting that this property is intrinsic to any functional metabolic network rather than a product of particular evolutionary histories. Indeed, the mechanistic explanation for this property in actual organisms (Fig. 3) holds in the random metabolic networks examined here. Specifically, the fraction of reactions exhibiting PC > TC is substantial (Fig. 4d) and $f_P$ is mostly lower than 1 (Fig. 4e). Furthermore, $f_P$ is generally more different from 1 than is $f_a$ in a $\log_{10}$ scale, because $|\log_{10}f_P| - |\log_{10}f_a|$ is largely positive (Fig. 4f).

Intriguingly, however, for 19 of the 20 new environments, $C_{RV}$ in the *E. coli* metabolic network exceeds the median $C_{RV}$ in the random networks (Fig. 4a). A similar but less obvious trend holds

for $C_{RI}$ (Fig. 4b). For 16 of the 20 new environments, $C_{RI}/C_{RV}$ from *E. coli* is smaller than the median $C_{RI}/C_{RV}$ of the random networks ($P = 0.012$, two-tailed binomial test; Fig. 4c). Hence, although both the *E. coli* metabolic network and random networks show a predominance of flux reversion, this phenomenon is more pronounced in *E. coli*. Mechanistically, this disparity is explainable at least qualitatively by our model in the previous section. Specifically, for 15 of the 20 new environments, the fraction of *E. coli* reactions with PC > TC exceeds the corresponding median fraction in random networks ($P = 0.021$, one-tailed binomial test; Fig. 4d). For all 20 new environments, $f_P$ of *E. coli* is lower than the median $f_P$ of random networks ($P = 9.5 \times 10^{-7}$, one-tailed binomial test; Fig. 4e). For 19 of the 20 new environments, $|\log_{10} f_P| - |\log_{10} f_a|$ for *E. coli* is larger than the corresponding median value for the random networks ($P = 2.0 \times 10^{-5}$, one-tailed binomial test; Fig. 4f). But, why is $f_P$ of *E. coli* lower than that of random networks? One potential explanation is that the composition and structure of the *E. coli* metabolic network have been evolutionarily optimized for growth in the glucose environment and/or related environments, while the same is not true for the random networks, which were only required to be viable in the glucose environment. As a result, when glucose is replaced with a new carbon source in a new environment, the fitness of *E. coli* drops substantially, but those of random networks may drop only mildly. Although the absolute fitness in the plastic stage may well be higher for *E. coli* than the random networks, the relative fitness, which $f_P$ is, is expected to be lower for *E. coli* than the random networks. Thus, the higher prevalence of flux reversion relative to reinforcement in *E. coli*

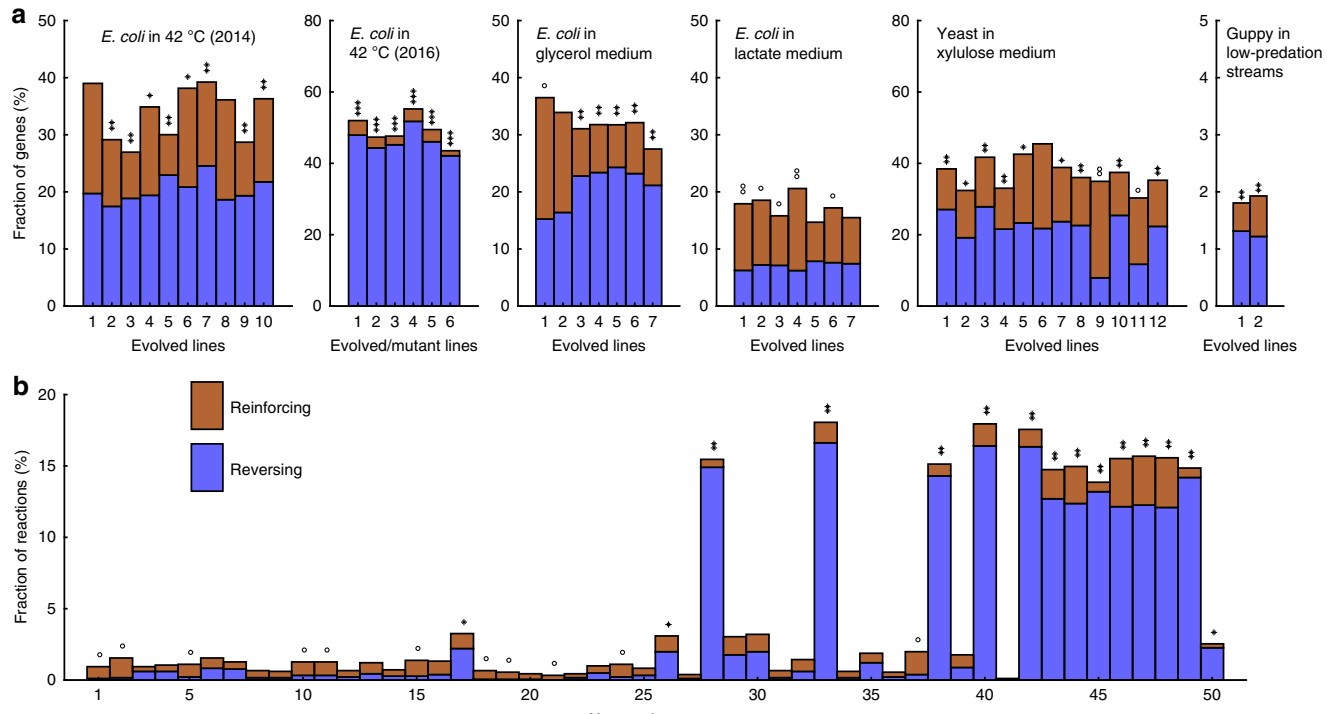

**Fig. 5** Fraction of reinforcing traits ($C_{RI}$) is no greater than that of reversing traits ($C_{RV}$) in adaptations even when the total change exceeds a preset cutoff. Traits satisfying $|L_a - L_o| > 0.2L_o$, $|L_p - L_o| > 0.2L_o$, and $|L_a - L_p| > 0.2L_o$ are classified into reinforcing and reversing traits based on whether the genetic and plastic changes are of the same direction or opposite directions. **a** Fractions of genes with reinforcing and reversing expression changes, respectively, in experimental evolution. Organisms as well as the new environments to which the organisms were adapting to are indicated. Each bar represents an adaptation. **b** Fractions of reactions with predicted reinforcing and reversing flux changes, respectively, in *E. coli*'s adaptations to 50 new environments from the glucose environment. In both panels, the equality in the fraction of reinforcing and reversing reactions is tested by a two-tailed binomial test. When $C_{RV} > C_{RI}$, $P$-values are indicated as follows: $*P < 0.05$; $**P < 10^{-10}$; $***P < 10^{-100}$; when $C_{RV} < C_{RI}$, $P$-values are indicated as follows: $^oP < 0.05$; $^{oo}P < 10^{-10}$; $^{ooo}P < 10^{-100}$

than random networks is likely a byproduct of stronger selection of *E. coli* compared with random networks in the original environment used in our adaptation analysis.

**Reversion is at least as common as reinforcement even for traits with appreciable TC.** In the foregoing analyses of transcriptomes (Fig. 1d) and fluxomes (Fig. 2a), we considered all traits exhibiting appreciable PCs and GCs. In comparative and evolutionary studies, however, phenotypes at stage p are typically inaccessible. As a result, comparative and evolutionary biologists usually focus on traits whose phenotypic values differ between stages o and a, despite that the other traits could have also experienced adaptive changes (from the values at stage p to those at stage a). To study if our foregoing findings apply to the traits that are the subject of most comparative and evolutionary biologists, we focus on a subset of traits above analyzed that satisfy the condition of $TC > 0.2L_o$. Of the 44 cases of experimental evolution, 33 showed $C_{RV} > C_{RI}$ ($P = 0.0013$, two-tailed binomial test), in 30 of which $C_{RV}$ significantly exceeds $C_{RI}$ (nominal $P < 0.05$; two-tailed binomial test; Fig. 5a). Of the 50 environmental adaptations of the *E. coli* metabolic network originating from the glucose environment, three cases had equal numbers of flux reversion and reinforcement. Among the remaining 47 cases, 22 showed more reversion than reinforcement, while 25 showed the opposite ($P = 0.77$, two-tailed binomial test; Fig. 5b). When $C_{RI}$ is significantly different from $C_{RV}$, 15 cases showed $C_{RV} < C_{RI}$ while 11 showed the opposite ($P = 0.70$, two-tailed binomial test; Fig. 5b). Hence, even among traits with $TC > 0.2L_o$, there is no evidence for significantly more reinforcement than reversion. Of note, in the above metabolic analysis, on average 139 reactions satisfied $TC > 0.2L_o$ per adaptation. Because all flux changes observed in the maximization of fitness are required and therefore are by definition beneficial, even the adaptation to a simple carbon source change apparently involves much more than a few reactions.

## Discussion

Using the transcriptome data collected in a total of 44 cases of six different experimental evolutionary adaptations of three species (*E. coli*, yeast, and guppy) and the computationally predicted fluxomes of *E. coli* in hundreds of different environmental adaptations, we showed that genetic adaptations to new environments overwhelmingly reverse, rather than reinforce plastic phenotypic changes. Our fluxome analyses have several caveats worth discussion. First, because MOMA minimizes the total squared flux difference from the original flux, PCs could have been underestimated, but this bias would only make our conclusion more conservative. Second, a bias could exist owing to potentially different accuracies of MOMA and FBA that are respectively used to predict plastic and genetic flux changes. To tackle this problem, we designed a MOMA-based algorithm to infer both PCs and GCs, but found the results to be qualitatively unchanged (Supplementary Fig. 4b). Third, we considered only single-carbon source environments in our analyses while the natural environments of *E. coli* can be much more complex. We thus simulated adaptations from the glucose environment to environments with mixed carbon sources (see Methods), but found our conclusion unaltered (Supplementary Fig. 6). Fourth, computational flux predictions by FBA and MOMA inevitably contain errors. But, the fact that our fluxome-based conclusion qualitatively match the transcriptome-based conclusion suggests that our fluxome analysis is reliable. Furthermore, some of our metabolic analyses are largely immune to flux prediction errors. For example, because the *E. coli* metabolic network and random metabolic networks were analyzed using the same method, their

difference discovered is unlikely explainable by flux prediction errors. As mentioned, our transcriptome analysis also has a potential shortcoming. Because the organisms were not fully adapted to the new environments at the end of experimental evolution, it is possible that a trait currently not considered to show reversion or reinforcement due to insufficient GC would show one of these two patterns if allowed to adapt further. However, because our results are robust to different cutoffs used ($0.05L_o$ to $0.5L_o$) in the definition of GCs (Supplementary Figs 2,3), our finding of the preponderance of expression level reversion is minimally impacted by this limitation. Another concern is that expression levels of many genes strongly correlate with organismal growth rate and may simply reflect the growth rate[40,41]; it is interesting to ask whether removing these genes would alter our result. Esquerre et al.[42] measured the transcriptomes of *E. coli* grown in a chemostat at four different rates. Using this data set, we defined a gene to be growth-rate-independent if its expression level does not monotonically increase or decrease with the growth rate, resulting in the assignment of 42% of genes as growth-rate-independent. Focusing on these genes in 30 cases of *E. coli* experimental evolution, we observed $C_{RV} > C_{RI}$ in 28 cases ($P = 8.7 \times 10^{-7}$, two-tailed binomial test), and $C_{RV}$ significantly exceeds $C_{RI}$ in each of these 28 cases (nominal $P < 0.05$; two-tailed binomial test; Supplementary Fig. 7). Thus, our finding also holds for growth-rate-independent genes.

In all analyses, we regarded phenotypic reinforcement as evidence for the stepping stone role of plasticity in adaptation and phenotypic reversion as evidence against this hypothesis[15]. One could argue that although reinforcement supports the hypothesis, reversion is not necessarily against the hypothesis. Specifically, if a PC moves the organismal phenotype closer to the optimum in the new environment but overshoots, the GC required to bring the phenotype to the optimum may be smaller than that in the absence of plasticity. To investigate this scenario, we considered all traits with PC and GC both larger than the cutoff as was done in the definition of reinforcement and reversion. We then regarded the PC of a trait as facilitating if GC < TC, or hindering if GC > TC. We respectively computed the fractions of traits with facilitating ($C_{FAC}$) and hindering PCs ($C_{HIN}$) in each adaptation. In 32 of the 44 cases of experimental evolution, $C_{HIN}$ exceeds $C_{FAC}$, demonstrating an overall preponderance of hindering plasticity ($P = 3.7 \times 10^{-3}$, two-tailed binomial test; Supplementary Fig. 8a). Furthermore, $C_{HIN}/C_{FAC}$ is likely underestimated in the above analysis, because the fact that adaptations to new environments had not ceased by the end of experimental evolution means that cases currently classified as facilitating can become hindering. This is because GC will probably rise in further adaptations while TC will either rise by at most the same amount as the increase in GC or reduce. For the adaptations of the *E. coli* metabolic network from the glucose environment to the 50 new environments, the above underestimation does not exist, and $C_{HIN}$ is found to exceed $C_{FAC}$ in every adaptation ($P = 1.8 \times 10^{-15}$, two-tailed binomial test; Supplementary Fig. 8b). Thus, the comparison between facilitating and hindering plasticity also refutes the hypothesis that plasticity is a stepping stone to adaptation.

It is also possible that the PC of a trait can move its phenotypic value to the optimal state in the new environment such that no GC is needed. But, we found that the fraction of traits with an appreciable PC ($PC > 0.2L_o$) but no appreciable GC ($GC < 0.2L_o$) in the transcriptome analysis of Fig. 1d is on average only 11%, which is likely an overestimate because the adaptation to the new environment may not have been completed in experimental evolution. The corresponding value is only 0.62% in the fluxome

analysis of Fig. 2a. Hence, even considering these cases does not alter our conclusion.

We provided evidence that the cause for the preponderance of phenotypic reversion is that, even with plasticity, organismal fitness drops precipitously after environmental shifts, but more or less recovers through subsequent evolution; such fitness trajectories dictate that many fitness-associated traits are drastically altered at the plastic stage but are then restored via adaptive evolution. Our model is consistent with the observation that stress response is frequently associated with growth cessation as well as reductions in the expression levels of growth-related genes and concentrations of central metabolites[43–45]. It is also consistent with the notion that genetic adaptation tends to rebalance the energy allocation in growth that is broken in stress response and that the physiological state of organisms after the rebalance in the new environment is similar to that in the original environment[16,18,44,46,47]. Together, these considerations suggest that plastic phenotypic changes in new environments represent emergency stress responses that may be important for organismal survival, but are otherwise not stepping stones for genetic adaptations to the new environments. The similar observation in functional random metabolic networks suggests that our conclusion is likely to be general to most functional systems regardless of the specific evolutionary histories of the systems.

Evolutionary biologists may contend that they are interested only in traits that differ between organisms living in different environments, because these traits have most likely experienced adaptive evolution. We showed that even for such traits (i.e., TC > $0.2L_o$), reinforcement is no more prevalent than reversion (Fig. 5), further refuting the stepping stone hypothesis. It is worth stressing, however, that a trait with TC < $0.2L_o$ may have also experienced adaptive evolution, because it could have a large PC reversed by a large GC that is beneficial. In other words, traits with similar values in stages o and a may have had cryptic adaptations unrevealed due to the lack of information about stage p. Hence, the observation that a trait looks similar among organisms living in different environments does not necessarily mean that it experiences no adaptive changes in organismal adaptations to their respective environments.

It is important to emphasize that our study focuses exclusively on adaptations to new environments that have not been experienced at least in the recent past. For those environments that have been (repeatedly) experienced by the organisms in the recent past, it is possible that mutations conferring plastic phenotypic changes that are beneficial in these environments have been fixed and there is no controversy that adaptive plasticity can evolve under this scenario.

The importance of plasticity in adaptation has also been discussed in theories of genetic assimilation[48] and accommodation[6], which refer to the evolutionary process by which a phenotype induced by an environmental stimulus becomes stably expressed even without the evoking environmental stimulus. Because the experimental evolution data analyzed do not contain information on the phenotypic plasticity of the organisms adapted to the new environment, our study cannot test genetic assimilation or accommodation. A related hypothesis that we did not test regarding the role of plasticity in adaptation is that upon an environmental shift, organisms with a relatively high plasticity adapt faster or are more likely to adapt than organisms with a relatively low plasticity. It would be interesting to test this hypothesis in the future when comparable organisms with contrasting levels of plasticity become available for experimental evolution studies.

Due to the limitation of the available data, our transcriptome and fluxome analyses focused primarily on unicellular microbes (with the exception of guppies). Compared with unicellulars, multicellulars are more complex because of differential gene expressions among cell types and because the biomass production rate of a cell type may not correlate well with organismal fitness. Therefore, it will be important to confirm the generality of our findings in the future when more data sets from multicellulars become available.

## Methods

**Gene expression analysis**. Transcriptome data sets from six experimental adaptations were acquired from five studies. For each replicate of each adaptation, the data included gene expression levels of ancestral organisms in the original environment (stage o), ancestral organisms in the new environment (stage p), and evolved organisms in the new environment (stage a). For each data set, we removed genes with any missing expression levels and then normalized gene expression levels such that the mean expression level of all genes is the same across all data sets.

The first data set came from the experimental evolution of *E. coli* K-12 MG1655 in a 42 °C medium with 10 replicates[17]. The authors performed RNA sequencing (RNA-seq) using (i) the ancestral line at 37 °C, (ii) ancestral line at 42 °C, and (iii) 10 parallelly evolved lines at 42 °C, and these data were respectively used to estimate the $L_o$, $L_p$, and $L_a$ of 4341 genes. All expression levels measured in FPKM were available in their Dataset S3.

The second data set came from the experimental evolution of *E. coli* B REL1206 in a 42 °C medium[18]. The authors performed RNA-seq using (i) the ancestral line at 37 °C, (ii) ancestral line at 42 °C, (iii) two evolved lines at 42 °C, and (iv) four lines each carrying a distinct adaptive mutation at 42 °C. We respectively used (i) to estimate $L_o$, (ii) to estimate $L_p$, and both (iii) and (iv) to estimate $L_a$ of 4202 genes. All expression levels measured by DESeq were provided by the authors.

The third and fourth data sets came from the experimental evolution of *E. coli* K-12 MG1655 in glycerol and lactate medium, respectively[16]. The authors used Affymetrix *E. coli* Antisense Genome Arrays to profile the transcriptome of (i) the ancestral line in glucose, (ii) ancestral line in glycerol, (iii) ancestral line in lactate, (iv) seven parallelly evolved lines in glycerol on day 21, (v) seven parallelly evolved lines in glycerol on day 44, (vi) seven parallelly evolved lines in lactate on day 20, and (vii) seven parallelly evolved lines in lactate on day 60. Each line has three replicates, except that profile (iii) has six replicates. We averaged gene expression levels across replicates for each line. For the adaptation to the glycerol medium, we respectively used (i) to estimate $L_o$, (ii) to estimate $L_p$, and (v) to estimate $L_a$. For the adaptation to the lactate medium, we respectively used (i) to estimate $L_o$, (iii) to estimate $L_p$, and (vii) to estimate $L_a$. Transcriptomes of (ii)–(vii) were downloaded from Gene Expression Omnibus (GEO) with the accession number GSE33147, whereas that of (i) was provided by the authors. In total, 3745 genes were considered.

The fifth data set came from the experimental evolution of 12 different strains of *S. cerevisiae* in a xylulose medium[19]. The authors performed RNA-seq using (i) 12 ancestral lines in a glucose medium, (ii) 12 ancestral lines in the xylulose medium, and (iii) 12 evolved lines in the xylulose medium. Each line has two replicates, and the averaged expression levels of the two replicates were used. We respectively used (i) to estimate $L_o$, (ii) to estimate $L_p$, and (iii) to estimate $L_a$ of 2235 genes. All expression levels in terms of UMI scoring normalized counts were downloaded from GEO with the accession number GSE76077.

The sixth data set came from the experimental evolution of *P. reticulata* guppies originating from streams with high numbers of cichlid predators (high predation (HP) environment) in cichlid-free streams (low predation (LP) environment)[15]. The authors performed RNA-seq of brain tissues from (i) guppies caught in HP, (ii) guppies caught in HP but reared in LP, and (iii) two populations of guppies in LP after experimental evolution. We respectively used (i) to estimate $L_o$, (ii) to estimate $L_p$, and (iii) to estimate $L_a$ of 37,493 genes. All expression levels in terms of TMM-normalized counts measured by edgeR were provided by the authors.

**Metabolic network analysis**. The SMBL file of the *E. coli* metabolic network model iAF1260[23] was downloaded from BiGG[49] and parsed by COBRA[50]. All linear and quadratic programming problems in this study were solved by the barrier method using Gurobi optimizer with MATLAB (method = 2). Numerical differences smaller than $10^{-4}$ were ignored in the analysis. The codes are available upon request.

We used FBA to estimate the fluxes of the *E. coli* network when it is fully adapted to an environment. FBA assumes a metabolic steady state and maximizes the rate of biomass production[20]. Mathematically, FBA is a linear programming question in the following form

$$\text{maximize } \mathbf{c}^T \mathbf{v}, \text{ subject to } \mathbf{Sv} = 0,$$
$$\text{and } \boldsymbol{\alpha} \leq \mathbf{v} \leq \boldsymbol{\beta},$$

where **v** is a vector of reaction fluxes that need to be optimized, $\mathbf{c}^T$ is a transposed vector describing the relative contributions of various metabolites to the cellular biomass, **S** is a matrix describing the stoichiometric relationships among metabolites in each reaction, $\boldsymbol{\alpha}$ is a vector describing the lower bound of each flux, and $\boldsymbol{\beta}$ is a vector describing the upper bound of each flux.

The model iAF1260 includes 258 exchange reactions, each of which allows the uptake of one carbon source. In the estimation of the fully adapted flux distribution in one environment, the uptake rate of the focal carbon source was set at 10 mmol g DW$^{-1}$ h$^{-1}$, which follows the setting in a previous study for a glucose-limited medium[23], while the uptake rates of other carbon sources were set at zero. The uptake rates of non-carbon sources such as water, oxygen, carbon dioxide, and ammonium were set as in the previous study[23]. Note that some reactions are simple diffusions between different cellular compartments. Because these reactions do not have dedicated enzymes and are not "mutable", we excluded them from the list of phenotypic traits considered. In total, 1811 reactions were considered.

We used MOMA to estimate plastic flux changes when E. coli is shifted from one environment to another[21]. The mathematical form of MOMA is

$$\text{minimize}(\mathbf{v} - \mathbf{v}_0)^2, \text{ subject to } \mathbf{Sv} = 0 \text{ and } \boldsymbol{\alpha} \le \mathbf{v} \le \boldsymbol{\beta},$$

where $\mathbf{v}$ is the vector of all reaction fluxes upon the environmental shift and is the variable to optimize, $\mathbf{v}_0$ is the vector of all reaction fluxes in the original environment and are predetermined using FBA. $\mathbf{S}$, $\boldsymbol{\alpha}$, and $\boldsymbol{\beta}$ are the same as described for FBA.

While MOMA was originally developed to predict metabolic fluxes upon gene deletions, MOMA developers discussed its potential applicability in predicting fluxes upon environmental shifts[21]. MOMA assumes that cells attempt to maintain the metabolic homeostasis as much as possible in the face of an unexperienced situation, may it be the loss of a reaction (due to gene deletion) or a change in the environment. In theory, an environmental change can be very similar to a gene deletion. For example, moving cells from the glucose medium to an unexperienced medium containing a different carbon source is equivalent to deleting genes for glucose transportation. Indeed, metabolic fluxes of E. coli respectively experimentally measured in lactate[51] and in galactose[52] correlate well with the fluxes predicted using MOMA (Supplementary Fig. 9). Therefore, MOMA is suitable for predicting plastic flux changes.

In the above investigation of MOMA performance, metabolic fluxes experimentally determined in the lactate medium were from Fig. 2a in Hua et al.[51]. The mapping from gene names to reaction names was based on the annotation in iAF1260. In total, nine genes (excluding lldD) were used. The relative flux of each of the nine reactions was calculated by the value underneath each box in Fig. 2a of Hua et al. divided by the lactate uptake rate in the lldD box. Metabolic fluxes experimentally determined in the galactose medium were from Fig. 1b and Supplementary Table 3 in Haverkorn van Rijsewijk et al.[52]. The mapping from gene names to reaction names was also based on the annotation in iAF1260. Note that we considered the flux measured for mae (MAL > PYR) as the combination of reactions ME1 and ME2 in iAF1260. In total, 26 measurements were used, and their relative fluxes were calculated by their values divided by the galactose uptake rate (2.17). For the relative fluxes predicted by MOMA, normalization was performed by using the estimated uptake rate of the corresponding carbon source in MOMA solutions.

To ensure that our results are not artifacts of different optimization accuracies of FBA and MOMA, we designed MOMA-b and used it to predict the fluxes in organisms adapted to new environments. In addition to having the same objective function and constraints as in MOMA, MOMA-b has a biomass constraint. Specifically, we set the biomass production rate in MOMA-b to be the same as what FBA predicts for organisms adapted to the new environment. The mathematical form of this new optimization question is

$$\text{minimize}(\mathbf{v} - \mathbf{v}_0)^2, \text{ subject to } \mathbf{Sv} = 0, \boldsymbol{\alpha} \le \mathbf{v} \le \boldsymbol{\beta}, \text{ and } \mathbf{c}^{\mathsf{T}}\mathbf{v} = b,$$

where the variables $\mathbf{v}$ and parameters $\mathbf{v}_0$, $\mathbf{S}$, $\boldsymbol{\alpha}$, and $\boldsymbol{\beta}$ are the same as described for MOMA, and $b$ is the FBA-predicted biomass production rate in the new environment. This optimization problem is still a quadratic programming problem and its solution can differ from that of FBA.

In addition to using single-carbon source environments, we followed a previous study[53] to generate 100 environments with multiple carbon sources. In each environment, we generated a random number $g$ from an exponential distribution with a mean of 0.1 for each of the 258 carbon sources. Here $g$ is the probability that the carbon source is available. The actual presence or absence of the carbon source is then determined stochastically using $g$. These random environments have a mean of 28 and a median of 21 carbon sources per environment.

**Data availability**. All relevant data are available from the corresponding author upon request.

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

### Acknowledgements

We thank members of the Zhang lab for valuable comments. This work was supported in part by the U.S. National Institutes of Health research grant GM103232 to J.Z.

### Author contributions

J.Z. conceived the project. W.-C.H. and J.Z. designed the research. W.-C.H. conducted the research and analyzed the data. W.-C.H. and J.Z. wrote the paper.

### Additional information

**Competing interests:** The authors declare no competing financial interests.

