## [Peer Review File · Nature Communications]

Reviewers' comments:

Reviewer #1 (Remarks to the Author):

This manuscript examines plastic and evolutionary changes in gene expression. The authors use data from different experimental evolution studies and compute metabolic fluxes from the transcriptome data. They find the initial plastic responses to a new environment are reversed by subsequent evolutionary changes. These results are significant for several reasons. First, by converting the transcriptome data into metabolic fluxes, they make important linkages from genotype to phenotype to fitness. Second, by examining multiple independent data sets they provide generality to their results.

Despite my enthusiasm for this study, there are still some important areas where the manuscript could be improved. I describe these and make suggestions for improvement below.

1) Conceptual Framework and Introduction

The second and third sentences of the introduction imply evolution only occurs through the fixation of beneficial mutations. While this may be the case with clones of *E. coli*, in natural populations rapid evolutionary changes are primarily derived from selection on standing genetic variation (see review by Barrett & Schluter 2008 *Trends in Ecol. & Evol.*). Thus, the opening introduction of a two phase evolutionary process needs to be rewritten to acknowledge that selection on standing variation can also result in evolutionary change.

While I generally agree with the "five considerations" presented for why it is beneficial to expand upon transcriptome data to consider metabolic fluxes, the authors should be more cautious in their presentation as the utility of metabolic flux analyses is questionable in complex eukaryotic organisms (e.g. Niklas et al. 2010). The metabolic fluxes measured in individual cells, do not necessarily scale up to changes in fitness at the whole organism scale for complex animals. Therefore, I would suggest caution in extrapolating too far in your considerations.

2) Methods and Experimental Design

I am not an expert in metabolic flux analysis, and thus cannot comment on the analyses. However, I know these pathways are well studied and trust the authors are competent in their approach.

3) Results

The results are compelling.

4) Discussion and Conclusions

I very much appreciated the critical approach taken in the discussion. While I was impressed by all the subsequent analyses, I was particularly interested in the scenario where plastic changes are in the same direction as evolution, but overshoot the optimum phenotype. While I liked the use of the terms reinforcement and reversion, I do not like the terms constructive and destructive to describe plasticity in this section. Please consider

alternatives terms, as constructive and destructive have a lot of negative connotations associated with them. Perhaps the terms “facilitating” vs. “constraining” could be used.

Reviewer #2 (Remarks to the Author):

The role that phenotypic plasticity plays in evolutionary adaptation is a subject of great interest and debate. One strategy that is increasingly employed to address this topic is to examine transcriptome data and look at how the expression of environmentally responsive genes changes over the course of adaptation. As the authors themselves recognize, there are some limitations to this approach—e.g., many, if not most, of the genes whose expression changes might not matter to this process and so inferences from these data could be misleading. To my knowledge, application of metabolic models represents a less utilized and potentially more informative way to address this problem.

Overall, this was an interesting manuscript that involved a number of rigorous and thoughtful analyses. The authors also did a good job of making key figures easier to understand using graphical abstractions. Further, I appreciated the balanced and comprehensive ways in which the authors approach their analyses and interpretation.

My more detailed comments on this paper are:

1. Sometimes the authors use language that is too strong. A good example of this might be the very first sentence of the abstract and the whole paper: ‘Organismal adaptation to a new environment typically starts with plastic phenotypic changes...’ The word ‘typically’ here seems inappropriate, as I don’t believe that this is known. There are other places where this logic could be applied. In this specific context, ‘can start’ might be more appropriate.
2. The placement of this paper and framing of hypotheses within the context of the broader literature seems incomplete or not specific enough. The authors seem to use their results to argue against the importance of plasticity to adaptation, emphasizing the idea that phenotypic plasticity might be an intermediate step in adaptation. However, it is my understanding that many, if not most, people interested in plasticity believe its importance may more so be in promoting genetic assimilation and genetic accommodation (note: The authors mainly cite these papers in the beginning of the paper, suggesting they want to connect their work to this literature). These processes occur when the environmental responsiveness of a plastic trait is reduced, enhanced, or even eliminated via genetic changes. These processes of genetic assimilation and genetic accommodation are different from the process that the authors describe in Fig. 1 and text, as they do not necessarily presume that a phenotype will change between steps ‘p’ and ‘a’ (see Fig, 1A). Rather, in genetic assimilation and genetic accommodation, it is the environmental responsiveness of the trait and/or variance in trait expression that would change. The authors need to be clearer regarding what they do and do not address about the role of plasticity in adaptation.

3. Connected to #2, the reinforcing vs. reversing dichotomy, as described and illustrated in Figure 1, is probably too simplified. The authors describe the reinforcing class as the situation where a genetic change results in a phenotypic change that extends and is in the same direction as the plastic response. However, wouldn't the reinforcing change often not extend the phenotypic change as in Fig. 1, but rather simply stabilize its expression? In this scenario, a different definition would be needed for reinforcement, as reinforcement could range from no change to a positive change in the same direction as the plastic change. How does this impact the analyses?

4. How to interpret the author's work regarding the role of plasticity in adaptation seems more ambiguous than the authors suggest. In the case of a plastic organismal phenotype being extended or stabilized through genetic changes, what are the expectations for how gene expression (or other systems features) should change? A plastic response could involve many genes showing a change in expression due to dysregulation of gene expression in a new condition. Only a subset of these genes might contribute to the plastic trait and so an important part of a process like genetic assimilation could be reversing the expression changes that do not matter for the adaptation while reinforcing the ones that do. This speaks to how the expectations for changes in expression during adaptation to a new environment are not well laid out and how analyzing changes in transcription is different from looking at an actual organismal phenotype that shows plasticity. The authors findings may be in line with what should be expected during processes like genetic assimilation and genetic accommodation.

Reviewer #3 (Remarks to the Author):

In this manuscript the authors present their work addressing the very interesting question of whether genetic changes tend to generally reinforce the plastic changes after a microbial organism is exposed to a new environment or not. To investigate this question the authors performed both: an analysis of existing expression data from several such experiments, as well as the simulation of metabolism in such circumstances using FBA and MOMA.

While the approach to the problem was in principle good and the manuscript is extremely well written there are several critical issues in both the data analysis as well as the simulation analysis which prevent the authors from reliably drawing the conclusion that most plastic changes are reversed genetically rather than reinforced.

Specifically, there are several issues in the analysis that would lead to an overestimation of the number of phenotype reversals.

In the gene expression analysis:

1) The way by which gene expression changes are classified as genetic is done by the authors by comparing gene expression levels between P and A states. However, many of these changes could arise from indirect regulatory effects, possibly arising from a single or only few mutations. How such changes are counted (as multiple or single genetic) changes could strongly affect the conclusion that most changes are reversed in the genetic evolution

phase.

2) In their discussion, the authors briefly dismiss stress response by saying "Our model is consistent with the observation that stress response is frequently associated with growth cessation...". The model may be consistent but more importantly the interpretation of the results is critically affected by such an effect. Since stress response is characterized by overexpression of normally unexpressed or lowly expressed genes (Weber et al 2005, Maurer et al 2005). After the organism genetically adapts to the new environment and the stress response ends, then one would expect to observe the expression of many genes return to their nominal levels, again here reversion would seem like a frequent case but only due to the confounding effect of the temporary stress response induced by the environmental change.

In the metabolic analysis:

3) It is a large leap to use flux predictions using MOMA as a proxy for expected gene expression in the P cells. Both predicted and real fluxes could be low while gene expression remained constant between simulated O and P states, if the flux was constrained due to upstream limited metabolite intake.

4) MOMA was developed for predicting the immediate impact on growth of genome modifications of a microbial organism, whether predictions made by MOMA hold true when using it to predict the impact on a changing environment is something that needs to be demonstrated before conclusions can be reliably drawn from such results.

MOMA makes predictions by finding the closest solution, in the new set of conditions (normally changed metabolic model due to gene KO, but in this case new environment), which match the previous FBA optimization solution in the previous condition. To illustrate why MOMA is unsuitable for predicting the immediate impact of metabolism in a new environment, consider the case of a glucose limited environment set to a maximum uptake of 5 units, following the authors' convention we call this state (O). If we consider now a "new" environment where the maximum uptake of glucose is set to 10 units and one applies MOMA, then the solution of MOMA will be identical to the solution of O, i.e.: the P solution will have maximally 5 units of glucose consumption and the same biomass yield as in the O environment. However, when using FBA, instead of MOMA, to predict the solution in the environment with maximum uptake of 10 units glucose, all fluxes will have been doubled including biomass yield. The bigger issue now arises when considering the differences in mass and energy between different metabolites. Take for example, maltose, a disaccharide composed of two glucose molecules. If as the new environment one supposes a maximal uptake of 5 units of maltose, then excluding the differences in transport reaction and conversion of maltose to glucose one arrives at almost the same solution as in the previous example. However, in this case the MOMA prediction would actually predict lower fluxes for most of the fluxes in the metabolic network (a result of trying to approximate the previous solution), resulting in the majority of changes being potentially considered reversals since, when doing the FBA optimization on maltose, the same situation with increased fluxes throughout would be apparent.

References:

1. Weber, H., Polen, T., Heuveling, J., Wendisch, V. F. & Hengge, R. Genome-wide analysis of the general stress response network in *Escherichia coli*: sigmaS-dependent genes, promoters, and sigma factor selectivity. *J. Bacteriol.* 187, 1591–603 (2005).
2. Maurer, L. M., Yohannes, E., Bondurant, S. S., Radmacher, M. & Slonczewski, J. L. pH regulates genes for flagellar motility, catabolism, and oxidative stress in *Escherichia coli* K-12. *J. Bacteriol.* 187, 304–19 (2005).

Response to the reviewers

We thank the three reviewers for their valuable comments, which have led to a substantial improvement of our manuscript. Below please find our point-to-point response.

Reviewer #1:

Comment 1

This manuscript examines plastic and evolutionary changes in gene expression. The authors use data from different experimental evolution studies and compute metabolic fluxes from the transcriptome data. They find the initial plastic responses to a new environment are reversed by subsequent evolutionary changes. These results are significant for several reasons. First, by converting the transcriptome data into metabolic fluxes, they make important linkages from genotype to phenotype to fitness. Second, by examining multiple independent data sets they provide generality to their results.

Despite my enthusiasm for this study, there are still some important areas where the manuscript could be improved. I describe these and make suggestions for improvement below.

Response

We thank the reviewer for recognizing the significance of our work. Please see below for our revisions in response to the suggestions.

Comment 2

1) Conceptual Framework and Introduction

The second and third sentences of the introduction imply evolution only occurs through the fixation of beneficial mutations. While this may be the case with clones of *E. coli*, in natural populations rapid evolutionary changes are primarily derived from selection on standing genetic variation (see review by Barrett & Schluter 2008 *Trends in Ecol. & Evol.*). Thus, the opening introduction of a two phase evolutionary process needs to be rewritten to acknowledge that selection on standing variation can also result in evolutionary change.

Response

We agree with the reviewer, and have modified the beginning of Introduction accordingly. Specifically, we now state that "Phenotypic adaptation to a new environment *can* comprise two phases (Fig. 1A)."

Comment 3

While I generally agree with the "five considerations" presented for why it is beneficial to expand upon transcriptome data to consider metabolic fluxes, the authors should be more cautious in their presentation as the utility of metabolic flux analyses is questionable in complex eukaryotic organisms (e.g. Niklas et al. 2010). The metabolic fluxes measured in individual cells, do not necessarily scale up to changes in fitness at the whole organism scale for complex animals. Therefore, I would suggest caution in extrapolating too far in your considerations.

Response

We agree with the reviewer and have added a paragraph to discuss the potential inapplicability of flux analysis in multicellular organisms (page 20, paragraph 2).

Comment 4

2) Methods and Experimental Design

I am not an expert in metabolic flux analysis, and thus cannot comment on the analyses. However, I know these pathways are well studied and trust the authors are competent in their approach.

Response

We thank the reviewer for the positive comment on our methodology.

Comment 5

3) Results

The results are compelling.

Response

We thank the reviewer for the positive evaluation of our results.

Comment 6

4) Discussion and Conclusions

I very much appreciated the critical approach taken in the discussion. While I was impressed by all the subsequent analyses, I was particularly interested in the scenario where plastic changes are in the same direction as evolution, but overshoot the optimum phenotype. While I liked the use of the terms reinforcement and reversion, I do not like the terms constructive and destructive to describe plasticity in this section. Please consider alternative terms, as constructive and destructive have a lot of negative connotations associated with them. Perhaps the terms “facilitating” vs. “constraining” could be used.

Response

Considering the reviewer’s comment, we changed the terms to “facilitating” and “hindering”. We think the term “constraining” is not accurate because overshooting does not constrain the subsequent genetic adaptation; it results in a larger phenotypic gap that needs to be bridged by genetic changes.

Reviewer #2:

Comment 1

The role that phenotypic plasticity plays in evolutionary adaptation is a subject of great interest and debate. One strategy that is increasingly employed to address this topic is to examine transcriptome data and look at how the expression of environmentally responsive genes changes over the course of adaptation. As the authors themselves recognize, there are some limitations to this approach—e.g., many, if not most, of the genes whose expression changes might not matter to this process and so inferences from these data could be misleading. To my knowledge, application of metabolic models represents a less utilized and potentially more informative way to address this problem.

Overall, this was an interesting manuscript that involved a number of rigorous and thoughtful analyses. The authors also did a good job of making key figures easier to understand using graphical abstractions. Further, I appreciated the balanced and comprehensive ways in which the authors approach their analyses and interpretation.

Response

We thank the reviewer for these positive evaluations.

Comment 2

My more detailed comments on this paper are:

Sometimes the authors use language that is too strong. A good example of this might be the very first sentence of the abstract and the whole paper: ‘Organismal adaptation to a new environment typically starts with plastic phenotypic changes...’ The word ‘typically’ here seems inappropriate, as I don’t believe that this is known. There are other places where this logic could be applied. In this specific context, ‘can start’ might be more appropriate.

Response

We appreciate these detailed suggestions and have made the following changes accordingly.

“Organismal adaptation to a new environment may ~~typically~~ start with plastic phenotypic changes followed by genetic changes.” (Beginning of Abstract)

“Phenotypic adaptation to a new environment can comprise two phases.” (Beginning of Introduction)

Comment 3

The placement of this paper and framing of hypotheses within the context of the broader literature seems incomplete or not specific enough. The authors seem to use their results to argue against the importance of plasticity to adaptation, emphasizing the idea that phenotypic plasticity might be an intermediate step in adaptation. However, it is my understanding that many, if not most, people interested in plasticity believe its importance may more so be in promoting genetic assimilation and genetic accommodation (note: The authors mainly cite these papers in the beginning of the paper, suggesting they want to connect their work to this literature). These processes occur when the environmental responsiveness of a plastic trait is reduced, enhanced, or even eliminated via genetic changes. These processes of genetic assimilation and genetic accommodation are different from the process that the authors describe in Fig. 1 and text, as they do not necessarily presume that a phenotype will change between steps ‘p’ and ‘a’ (see Fig, 1A). Rather, in genetic assimilation and genetic accommodation, it is the environmental responsiveness of the trait and/or variance in trait expression that would change. The authors need to be clearer regarding what they do and do not address about the role of plasticity in adaptation.

Response

Following the suggestion, we have added the following discussion on how our results are related to genetic assimilation and accommodation (page 19, paragraph 4).

"The importance of plasticity in adaption has also been discussed in theories of genetic assimilation⁴⁸ and accommodation⁶, which refer to the evolutionary process by which a phenotype induced by an environmental stimulus becomes stably expressed even without the evoking environmental stimulus. Because the experimental evolution data analyzed do not contain information on the phenotypic plasticity of the organisms adapted to the new environment, our study cannot test genetic assimilation or accommodation. A related hypothesis that we did not test regarding the role of plasticity in adaptation is that, upon an environmental shift, organisms with a relatively high plasticity adapt faster or are more likely to adapt than organisms with a relatively low plasticity. It would be interesting to test this hypothesis in the future when comparable organisms with contrasting levels of plasticity become available for experimental evolution studies."

Comment 4

3. Connected to #2, the reinforcing vs. reversing dichotomy, as described and illustrated in Figure 1, is probably too simplified. The authors describe the reinforcing class as the situation where a genetic change results in a phenotypic change that extends and is in the same direction as the plastic response. However, wouldn't the reinforcing change often not extend the phenotypic change as in Fig. 1, but rather simply stabilize its expression? In this scenario, a different definition would be needed for reinforcement, as reinforcement could range from no change to a positive change in the same direction as the plastic change. How does this impact the analyses?

Response

The reviewer asks how often a plastic change is not followed by a genetic change. In the transcriptome analysis of experimental evolution, we found that on average only 11% of genes show a plastic change without genetic change. This is likely an overestimate because the adaptation to the new environment may not have been completed in experimental evolution. In the fluxome analysis, only 0.62% of fluxes show a plastic change without genetic change. Together, these results indicate that our conclusion is unaltered even when genes/fluxes with plastic changes only are considered. We added this information to the manuscript (page 18, paragraph 2).

Comment 5

4. How to interpret the author's work regarding the role of plasticity in adaptation seems more ambiguous than the authors suggest. In the case of a plastic organismal phenotype being extended or stabilized through genetic changes, what are the expectations for how gene expression (or other systems features) should change? A plastic response could involve many genes showing a change in expression due to dysregulation of gene expression in a new condition. Only a subset of these genes might contribute to the plastic trait and so an important part of a process like genetic assimilation could be reversing the expression changes that do not matter for the adaptation while reinforcing the ones that do. This speaks to how the expectations for changes in expression during adaptation to a new environment are not well laid out and how analyzing changes in transcription is different from looking at an actual organismal phenotype that shows plasticity. The authors findings may be in LINE with what should be expected during processes like genetic assimilation and genetic accommodation.

Response

In the case of transcriptome analysis, it is true that not every gene expression change may be adaptive or even relevant to adaptation. However, in the case of fluxome analysis, every flux change is adaptive because all flux changes identified are required for fitness maximization. This contrast is discussed in the section on the benefits of the fluxome analysis over transcriptome analysis (page 5, paragraph 1). Regarding the relation to genetic assimilation and accommodation, please refer to our response to Comment 3.

Reviewer #3:

Comment 1

In this manuscript the authors present their work addressing the very interesting question of whether genetic changes tend to generally reinforce the plastic changes after a microbial organism is exposed to a new environment or not. To investigate this question the authors performed both: an analysis of existing expression data from several such experiments, as well as the simulation of metabolism in such circumstances using FBA and MOMA.

While the approach to the problem was in principle good and the manuscript is extremely well written there are several critical issues in both the data analysis as well as the simulation analysis which prevent the authors from reliably drawing the conclusion that most plastic changes are reversed genetically rather than reinforced.

Specifically, there are several issues in the analysis that would lead to an overestimation of the number of phenotype reversals.

Response

We thank the reviewer for the overall positive evaluation. See below for our detailed response.

Comment 2

In the gene expression analysis:

1) The way by which gene expression changes are classified as genetic is done by the authors by comparing gene expression levels between P and A states. However, many of these changes could arise from indirect regulatory effects, possibly arising from a single or only few mutations. How such changes are counted (as multiple or single genetic) changes could strongly affect the conclusion that most changes are reversed in the genetic evolution phase.

Response

The reviewer is right that one mutation could affect the expressions of multiple genes, but our analysis focuses on the number of phenotypic traits (i.e., number of genes/reactions whose expression/flux levels are altered) rather than the number of mutations, because plastic changes by definition refer to phenotypic changes. We have edited the manuscript to make this point clearer (page 6, paragraph 2).

Comment 3

2) In their discussion, the authors briefly dismiss stress response by saying “Our model is consistent with the observation that stress response is frequently associated with growth cessation...”. The model may be consistent but more importantly the interpretation of the results is critically affected by such an effect. Since stress response is characterized by overexpression of normally unexpressed or lowly expressed genes (Weber et al 2005, Maurer et al 2005). After the organism genetically adapts to the new environment and the stress response ends, then one would expect to observe the expression of many genes return to their nominal levels, again here reversion would seem like a frequent case but only due to the confounding effect of the temporary stress response induced by the environmental change.

Response

We did not dismiss stress response. Rather, we suggest that plasticity at least in part reflects stress response. We have edited our manuscript to make our point clearer (end of page 18).

Comment 4

In the metabolic analysis:

3) It is a large leap to use flux predictions using MOMA as a proxy for expected gene expression in the P cells. Both predicted and real fluxes could be low while gene expression remained constant between simulated O and P states, if the flux was constrained due to upstream limited metabolite intake.

Response

We did not use fluxes as a proxy for expression levels. We used fluxes as a different set of traits. We have edited the manuscript to make this distinction clearer (page 7, paragraph 2).

Comment 5

4) MOMA was developed for predicting the immediate impact on growth of genome modifications of a microbial organism, whether predictions made by MOMA hold true when using it to predict the impact on a changing environment is something that needs to be demonstrated before conclusions can be reliably drawn from such results.

MOMA makes predictions by finding the closest solution, in the new set of conditions (normally changed metabolic model due to gene KO, but in this case new environment), which match the previous FBA optimization solution in the previous condition. To illustrate why MOMA is unsuitable for predicting the immediate impact of metabolism in a new environment, consider the case of a glucose limited environment set to a maximum uptake of 5 units, following the authors' convention we call this state (O). If we consider now a "new" environment where the maximum uptake of glucose is set to 10 units and one applies MOMA, then the solution of MOMA will be identical to the solution of O, i.e.: the P solution will have maximally 5 units of glucose consumption and the same biomass yield as in the O environment. However, when using FBA, instead of MOMA, to predict the solution in the environment with maximum uptake of 10 units glucose, all fluxes will have been doubled including biomass yield. The bigger issue now arises when considering the differences in mass and energy between different metabolites. Take for example, maltose, a disaccharide composed of two glucose molecules. If as the new environment one supposes a maximal uptake of 5 units of maltose, then excluding the differences in transport reaction and conversion of maltose to glucose one arrives at almost the same solution as in the previous example. However, in this case the MOMA prediction would actually predict lower fluxes for most of the fluxes in the metabolic network (a result of trying to approximate the previous solution), resulting in the majority of changes being potentially considered reversals since, when doing the FBA optimization on maltose, the same situation with increased fluxes throughout would be apparent.

Response

We thank the reviewer for this important comment. In principle, the effect of gene deletion is analogous to that of changing the current environment to an unexperienced environment. In an unexperienced environment, cells do not know the optimal solution and should try to maintain the original fluxes as much as possible, as in the case of gene deletion. We have now added analyses demonstrating the good performance of MOMA predictions upon two environmental changes (Fig. S9). If glucose is switched to maltose and if the maltose environment is an unexperienced environment (as we study adaptations to unexperienced environments in this work), we expect cells to behave as what MOMA predicts (i.e., very low maltose transportation flux and low growth). We have added the following paragraph in Methods (page 24, paragraph 2).

*"While MOMA was originally developed to predict metabolic fluxes upon gene deletions, MOMA developers discussed its potential applicability in predicting fluxes upon environmental shifts²¹. MOMA assumes that cells attempt to maintain the metabolic homeostasis as much as possible in the face of an unexperienced situation, may it be the loss of a reaction (due to gene deletion) or a change in the environment. In theory, an environmental change can be very similar to a gene deletion. For example, moving cells from the glucose medium to an unexperienced environment that contains a different carbon source is equivalent to deleting genes for glucose transportation. Indeed, metabolic fluxes of *E. coli* respectively experimentally measured in lactate⁵¹ and in galactose⁵² correlate well with the fluxes predicted using MOMA (Fig. S9). Therefore, MOMA is suitable for predicting plastic flux changes."*

Reviewers' comments:

Reviewer #2 (Remarks to the Author):

This is an interesting manuscript that has been improved through revision. I had two minor comments:

1. On p6 near lines 130 to 133. It might be of value to explicitly state that the time between stage 'o' and stage 'p' was too short for mutations to have arisen and caused observed gene expression changes.
2. I found the section from lines 219 to 240 a bit dense and generally thought this key part of the paper could be made a bit easier to read. A specific suggestion regards line 221 and the sentences immediately following it. The authors state that their analyses suggest a general underlying mechanism, which is not proposed until a few sentences later. It might be more straightforward to provide the mechanism immediately following that initial statement and then describe the equations supporting it.

Reviewer #3 (Remarks to the Author):

Comment 2

In the gene expression analysis:

- 1) The way by which gene expression changes are classified as genetic is done by the authors by comparing gene expression levels between P and A states. However, many of these changes could arise from indirect regulatory effects, possibly arising from a single or only few mutations. How such changes are counted (as multiple or single genetic) changes could strongly affect the conclusion that most changes are reversed in the genetic evolution phase.

Author response

The reviewer is right that one mutation could affect the expressions of multiple genes, but our analysis focuses on the number of phenotypic traits (i.e., number of genes/reactions whose expression/flux levels are altered) rather than the number of mutations, because plastic changes by definition refer to phenotypic changes. We have edited the manuscript to make this point clearer (page 6, paragraph 2).

Reviewer response

The question the authors want to address is whether genetic mutations reinforce or reverse plastic phenotypic changes. It seems to me that to answer this question the authors must consider only the more direct effects of the adaptive mutations excluding any indirect effects due to preprogrammed responses which constitute the preexisting phenotype of the organism and did not change in the adaptation phase. In other words, if we consider stress

response as the main factor in the observed gene expression changes in the P state, the adaptive mutations most likely did not remove the stress response behaviour, but instead resulted in an increased expression of a pathway that led to stress response being reduced. If pathway upregulation is a common occurrence by which organisms adapt, then the authors may arrive at the opposite conclusion, that the phenotype changes more directly induced by the adaptive mutations generally reinforce or maintain the plastic changes brought on by stress response.

A suggestion as to how the authors could proceed is to compare the gene expression changes between the O and A states, and do the same analysis only on this subset. This will allow the authors to identify those changes that exclude at least a large part of changes induced due to stress response and consider the gene expression changes resulting more directly from adaptive mutations.

Comment 3

2) In their discussion, the authors briefly dismiss stress response by saying "Our model is consistent with the observation that stress response is frequently associated with growth cessation...". The model may be consistent but more importantly the interpretation of the results is critically affected by such an effect. Since stress response is characterized by overexpression of normally unexpressed or lowly expressed genes (Weber et al 2005, Maurer et al 2005). After the organism genetically adapts to the new environment and the stress response ends, then one would expect to observe the expression of many genes return to their nominal levels, again here reversion would seem like a frequent case but only due to the confounding effect of the temporary stress response induced by the environmental change.

Author response

We did not dismiss stress response. Rather, we suggest that plasticity at least in part reflects stress response. We have edited our manuscript to make our point clearer (end of page 18).

Reviewer response

Already addressed in previous comment.

Comment 4

In the metabolic analysis:

3) It is a large leap to use flux predictions using MOMA as a proxy for expected gene expression in the P cells. Both predicted and real fluxes could be low while gene expression remained constant between simulated O and P states, if the flux was constrained due to upstream limited metabolite intake.

Authors response

We did not use fluxes as a proxy for expression levels. We used fluxes as a different set of traits. We have edited the manuscript to make this distinction clearer (page 7, paragraph 2).

Reviewer response

While one can in principle label virtually any physical (or behavioural) property of an organism a phenotype, by definition only properties that are affected by the genotype can be labeled as phenotypes. As an example, temperature in an organism that does not regulate its temperature can not be considered a phenotype as it results simply due to thermal equilibrium with the environment.

It seems to me that many metabolic flux changes may not be considered phenotype changes since they can change simply due to physico-chemical constraints resulting from upstream or downstream changes in metabolite concentrations without any corresponding changes in gene expression or metabolic regulation that would have been encoded in the genotype.

Here I suggest the authors take a similar approach to the approach suggested for the gene expression analysis in my reply in comment 2, restricting the analysis to changes found between the O and A states. This does not directly solve the problem of considering fluxes to be phenotypes, i.e.: properties under the influence of an organisms phenotype. But for at least some fraction of the flux changes (between O and A) it is reasonable to expect some underlying genotype changes.

Comment 5

4) MOMA was developed for predicting the immediate impact on growth of genome modifications of a microbial organism, whether predictions made by MOMA hold true when using it to predict the impact on a changing environment is something that needs to be demonstrated before conclusions can be reliably drawn from such results.

MOMA makes predictions by finding the closest solution, in the new set of conditions (normally changed metabolic model due to gene KO, but in this case new environment), which match the previous FBA optimization solution in the previous condition. To illustrate why MOMA is unsuitable for predicting the immediate impact of metabolism in a new environment, consider the case of a glucose limited environment set to a maximum uptake of 5 units, following the authors' convention we call this state (O). If we consider now a "new" environment where the maximum uptake of glucose is set to 10 units and one applies MOMA, then the solution of MOMA will be identical to the solution of O, i.e.: the P solution will have maximally 5 units of glucose consumption and the same biomass yield as in the O environment. However, when using FBA, instead of MOMA, to predict the solution in the environment with maximum uptake of 10 units glucose, all fluxes will have been doubled

including biomass yield. The bigger issue now arises when considering the differences in mass and energy between different metabolites. Take for example, maltose, a disaccharide composed of two glucose molecules. If as the new environment one supposes a maximal uptake of 5 units of maltose, then excluding the differences in transport reaction and conversion of maltose to glucose one arrives at almost the same solution as in the previous example. However, in this case the MOMA prediction would actually predict lower fluxes for most of the fluxes in the metabolic network (a result of trying to approximate the previous solution), resulting in the majority of changes being potentially considered reversals since, when doing the FBA optimization on maltose, the same situation with increased fluxes throughout would be apparent.

Authors response

We thank the reviewer for this important comment. In principle, the effect of gene deletion is analogous to that of changing the current environment to an unexperienced environment. In an unexperienced environment, cells do not know the optimal solution and should try to maintain the original fluxes as much as possible, as in the case of gene deletion. We have now added analyses demonstrating the good performance of MOMA predictions upon two environmental changes (Fig. S9). If glucose is switched to maltose and if the maltose environment is an unexperienced environment (as we study adaptations to unexperienced environments in this work), we expect cells to behave as what MOMA predicts (i.e., very low maltose transportation flux and low growth). We have added the following paragraph in Methods (page 24, paragraph 2).

"While MOMA was originally developed to predict metabolic fluxes upon gene deletions, MOMA developers discussed its potential applicability in predicting fluxes upon environmental shifts²¹. MOMA assumes that cells attempt to maintain the metabolic homeostasis as much as possible in the face of an unexperienced situation, may it be the loss of a reaction (due to gene deletion) or a change in the environment. In theory, an environmental change can be very similar to a gene deletion. For example, moving cells from the glucose medium to an unexperienced environment that contains a different carbon source is equivalent to deleting genes for glucose transportation. Indeed, metabolic fluxes of *E. coli* respectively experimentally measured in lactate⁵¹ and in galactose⁵² correlate well with the fluxes predicted using MOMA (Fig. S9). Therefore, MOMA is suitable for predicting plastic flux changes."

Reviewer response

I appreciate the effort the authors have put into making a convincing case for the use of MOMA in this type of predictions. While there are clearly some cases in which MOMA would give results that likely would not match experimental observations such as when doubling the maximum uptake rate of a limiting metabolite (glucose), I am convinced that such an effect will not fundamentally change the results of the authors in the cases considered here.

As their work may be cited in the future exactly to support the use of MOMA in this type of

application, I would ask the authors to describe in more detail the data they used from the two publications cited and how they arrived at the results for Fig. S9.

RESPONSE TO REVIEWERS

Reviewer #2:

Comment 1

This is an interesting manuscript that has been improved through revision. I had two minor comments:

On p6 near lines 130 to 133. It might be of value to explicitly state that the time between stage 'o' and stage 'p' was too short for mutations to have arisen and caused observed gene expression changes.

Response

Modified as suggested.

Comment 2

I found the section from lines 219 to 240 a bit dense and generally thought this key part of the paper could be made a bit easier to read. A specific suggestion regards line 221 and the sentences immediately following it. The authors state that their analyses suggest a general underlying mechanism, which is not proposed until a few sentences later. It might be more straightforward to provide the mechanism immediately following that initial statement and then describe the equations supporting it.

Response

Modified as suggested.

Reviewer #3:

Comment 1

Comment 2

In the gene expression analysis:

1) The way by which gene expression changes are classified as genetic is done by the authors by comparing gene expression levels between P and A states. However, many of these changes could arise from indirect regulatory effects, possibly arising from a single or only few mutations. How such changes are counted (as multiple or single genetic) changes could strongly affect the conclusion that most changes are reversed in the genetic evolution phase.

Author response

The reviewer is right that one mutation could affect the expressions of multiple genes, but our analysis focuses on the number of phenotypic traits (i.e., number of genes/reactions whose expression/flux levels are altered) rather than the number of mutations, because plastic changes by definition refer to phenotypic changes. We have edited the manuscript to make this point clearer (page 6, paragraph 2).

Reviewer response

The question the authors want to address is whether genetic mutations reinforce or reverse plastic phenotypic changes. It seems to me that to answer this question the authors must consider only the more direct effects of the adaptive mutations excluding any indirect effects due to preprogrammed responses which constitute the preexisting phenotype of the organism and did not change in the adaptation phase. In other words, if we consider stress response as the main factor in the observed gene expression changes in the P state, the adaptive mutations most likely did not remove the stress response behaviour, but instead resulted in an increased expression of a pathway that led to stress response being reduced. If pathway upregulation is a common occurrence by which organisms adapt, then the authors may arrive at the opposite conclusion, that the phenotype changes more directly induced by the adaptive mutations generally reinforce or maintain the plastic changes brought on by stress response.

A suggestion as to how the authors could proceed is to compare the gene expression changes between the O and A states, and do the same analysis only on this subset. This will allow the authors to identify those changes that exclude at least a large part of changes induced due to stress response and consider the gene expression changes resulting more directly from adaptive mutations.

Comment 3

2) In their discussion, the authors briefly dismiss stress response by saying “Our model is consistent with the observation that stress response is frequently associated with growth cessation...”. The model may be consistent but more importantly the interpretation of the results is critically affected by such an effect. Since stress response is characterized by overexpression of normally unexpressed or lowly expressed genes (Weber et al 2005, Maurer et al 2005). After the organism genetically adapts to the new environment and the stress response ends, then one would expect to observe the expression of many genes return to their nominal levels, again here reversion would seem like a frequent case but only due to the confounding effect of the temporary stress response induced by the environmental change.

Author response

We did not dismiss stress response. Rather, we suggest that plasticity at least in part reflects stress response. We have edited our manuscript to make our point clearer (end of page 18).

Reviewer response

Already addressed in previous comment.

Comment 4

In the metabolic analysis:

3) It is a large leap to use flux predictions using MOMA as a proxy for expected gene expression in the P cells. Both predicted and real fluxes could be low while gene expression remained constant between simulated O and P states, if the flux was constrained due to upstream limited metabolite intake.

Authors response

We did not use fluxes as a proxy for expression levels. We used fluxes as a different set of traits. We have edited the manuscript to make this distinction clearer (page 7, paragraph 2).

Reviewer response

While one can in principle label virtually any physical (or behavioural) property of an organism a phenotype, by definition only properties that are affected by the genotype can be labeled as phenotypes. As an example, temperature in an organism that does not regulate its temperature can not be considered a phenotype as it results simply due to thermal equilibrium with the environment.

It seems to me that many metabolic flux changes may not be considered phenotype changes since they can change simply due to physico-chemical constraints resulting from upstream or downstream changes in metabolite concentrations without any corresponding changes in gene expression or metabolic regulation that would have been encoded in the genotype.

Here I suggest the authors take a similar approach to the approach suggested for the gene expression analysis in my reply in comment 2, restricting the analysis to changes found between the O and A states. This does not directly solve the problem of considering fluxes to be phenotypes, i.e.: properties under the influence of an organisms phenotype. But for at least some fraction of the flux changes (between O and A) it is reasonable to expect some underlying genotype changes.

Response

We thank the reviewer for these comments and for suggesting a solution. Following the suggestion, we analyze only the traits with an appreciable change in phenotype between stages *o* and *a*. Again, we do not find more reinforcement than reversion. See the section “Reversion is at least as common as reinforcement even for traits with appreciable *TC*” on pages 14-15 and Fig. 5.

Comment 2

Comment 5

4) MOMA was developed for predicting the immediate impact on growth of genome modifications of a microbial organism, whether predictions made by MOMA hold true when using it to predict the impact on a changing environment is something that needs to be demonstrated before conclusions can be reliably drawn from such results.

MOMA makes predictions by finding the closest solution, in the new set of conditions (normally changed metabolic model due to gene KO, but in this case new environment), which match the previous FBA optimization solution in the previous condition. To illustrate why MOMA is unsuitable for predicting the immediate impact of metabolism in a new environment, consider the case of a glucose limited environment set to a maximum uptake of 5 units, following the authors' convention we call this state (O). If we consider now a “new” environment where the maximum uptake of glucose is set to 10 units and one applies MOMA, then the solution of

MOMA will be identical to the solution of O, i.e.: the P solution will have maximally 5 units of glucose consumption and the same biomass yield as in the O environment. However, when using FBA, instead of MOMA, to predict the solution in the environment with maximum uptake of 10 units glucose, all fluxes will have been doubled including biomass yield. The bigger issue now arises when considering the differences in mass and energy between different metabolites. Take for example, maltose, a disaccharide composed of two glucose molecules. If as the new environment one supposes a maximal uptake of 5 units of maltose, then excluding the differences in transport reaction and conversion of maltose to glucose one arrives at almost the same solution as in the previous example. However, in this case the MOMA prediction would actually predict lower fluxes for most of the fluxes in the metabolic network (a result of trying to approximate the previous solution), resulting in the majority of changes being potentially considered reversals since, when doing the FBA optimization on maltose, the same situation with increased fluxes throughout would be apparent.

Authors response

We thank the reviewer for this important comment. In principle, the effect of gene deletion is analogous to that of changing the current environment to an unexperienced environment. In an unexperienced environment, cells do not know the optimal solution and should try to maintain the original fluxes as much as possible, as in the case of gene deletion. We have now added analyses demonstrating the good performance of MOMA predictions upon two environmental changes (Fig. S9). If glucose is switched to maltose and if the maltose environment is an unexperienced environment (as we study adaptations to unexperienced environments in this work), we expect cells to behave as what MOMA predicts (i.e., very low maltose transportation flux and low growth). We have added the following paragraph in Methods (page 24, paragraph 2).

"While MOMA was originally developed to predict metabolic fluxes upon gene deletions, MOMA developers discussed its potential applicability in predicting fluxes upon environmental shifts²¹. MOMA assumes that cells attempt to maintain the metabolic homeostasis as much as possible in the face of an unexperienced situation, may it be the loss of a reaction (due to gene deletion) or a change in the environment. In theory, an environmental change can be very similar to a gene deletion. For example, moving cells from the glucose medium to an unexperienced environment that contains a different carbon source is equivalent to deleting genes for glucose transportation. Indeed, metabolic fluxes of *E. coli* respectively experimentally measured in lactate⁵¹ and in galactose⁵² correlate well with the fluxes predicted using MOMA (Fig. S9). Therefore, MOMA is suitable for predicting plastic flux changes."

Reviewer response

I appreciate the effort the authors have put into making a convincing case for the use of MOMA in this type of predictions. While there are clearly some cases in which MOMA would give results that likely would not match experimental observations such as when doubling the maximum uptake rate of a limiting metabolite (glucose), I am convinced that such an effect will not fundamentally change the results of the authors in the cases considered here. As their work may be cited in the future exactly to support the use of MOMA in this type of

application, I would ask the authors to describe in more detail the data they used from the two publications cited and how they arrived at the results for Fig. S9.

Response

We agree and have added details on how we acquired and analyzed the data (page 24, paragraph 3).

REVIEWERS' COMMENTS:

Reviewer #3 (Remarks to the Author):

In the revised manuscript the authors have made a substantial effort and include now several analysis that address all the concerns raised. I believe the latest results definitely make this work even more interesting.